# Vasoactive Properties of a Cocoa Shell Extract: Mechanism of Action and Effect on Endothelial Dysfunction in Aged Rats

**DOI:** 10.3390/antiox11020429

**Published:** 2022-02-21

**Authors:** Pilar Rodríguez-Rodríguez, Kendal Ragusky, Sophida Phuthong, Santiago Ruvira, David Ramiro-Cortijo, Silvia Cañas, Miguel Rebollo-Hernanz, María Dolores Morales, Ángel L. López de Pablo, María A. Martín-Cabrejas, Silvia M. Arribas

**Affiliations:** 1Department of Physiology, Faculty of Medicine, Universidad Autónoma de Madrid, C/Arzobispo Morcillo 2, 28029 Madrid, Spain; pilar.rodriguezr@uam.es (P.R.-R.); kendal.ragusky@estudiante.uam.es (K.R.); santiago.ruvira@estudiante.uam.es (S.R.); david.ramiro@uam.es (D.R.-C.); angel.lopezdepablo@uam.es (Á.L.L.d.P.); 2Food, Oxidative Stress and Cardiovascular Health (FOSCH) Multidisciplinary Research Team, Universidad Autónoma de Madrid, Ciudad Universitaria de Cantoblanco, 28049 Madrid, Spain; silvia.cannas@uam.es (S.C.); miguel.rebollo@uam.es (M.R.-H.); maria.martin@uam.es (M.A.M.-C.); 3Department of Physiology, Faculty of Medicine, Khon Kaen University, Khon Kaen 40002, Thailand; sophiph@kku.ac.th; 4PhD Programme in Pharmacology and Physiology, Doctoral School, Universidad Autónoma de Madrid, 28049 Madrid, Spain; 5Department of Agricultural Chemistry and Food Science, Faculty of Science, Universidad Autónoma de Madrid, Ciudad Universitaria de Cantoblanco, 28049 Madrid, Spain; 6Institute of Food Science Research, CIAL (UAM-CSIC), Universidad Autonoma de Madrid, C/Nicolás Cabrera, 9, 28049 Madrid, Spain; 7Confocal Microscopy Service (SiDI), Faculty of Medicine, Universidad Autonoma de Madrid, C/Arzobispo Morcillo 2, 28029 Madrid, Spain; confocal.sidi@uam.es

**Keywords:** cocoa shell, caffeine, endothelial dysfunction, by-products revalorization, nitric oxide, protocatechuic acid, theobromine, superoxide anion

## Abstract

Cocoa has cardiovascular beneficial effects related to its content of antioxidant phytochemicals. Cocoa manufacturing produces large amounts of waste, but some by-products may be used as ingredients with health-promoting potential. We aimed to investigate the vasoactive actions of an extract from cocoa shell (CSE), a by-product containing theobromine (TH), caffeine (CAF) and protocatechuic acid (PCA) as major phytochemicals. In carotid and iliac arteries from 5-month and 15-month-old rats, we investigated CSE vasoactive properties, mechanism of action, and the capacity of CSE, TH, CAF and PCA to improve age-induced endothelial dysfunction. Vascular function was evaluated using isometric tension recording and superoxide anion production by dihydroethidium (DHE) staining and confocal microscopy. CSE caused endothelium-dependent vasorelaxation, blocked by L-NAME, but not indomethacin, regardless of sex, age, or vessel type. CSE maximal responses and EC_50_ were significantly lower compared to acetylcholine (ACh). Arterial preincubation with CSE, TH, CAF or PCA, significantly reduced the number of vascular DHE-positive cells. Compared to adult males, iliac arteries from aged males exhibited reduced ACh concentration-dependent vasodilatation but larger CSE responses. In iliac arteries from aged male and female rats, preincubation with 10^−4^ M CSE and PCA, but not TH or CAF, improved ACh-relaxations. In conclusion, CSE has vasodilatory properties associated with increased nitric oxide bioavailability, related to its antioxidant phytochemicals, being particularly relevant PCA. Therefore, CSE is a potential food ingredient for diseases related to endothelial dysfunction.

## 1. Introduction

According to the World Health Organization, cardiovascular disease (CVD) is the leading cause of morbidity and mortality worldwide, accounting for nearly 18 million deaths annually, placing strain on both the individual and public health systems at large [1]. One of the main alterations involved in the pathogenesis of CVD is endothelial dysfunction. Once thought to be a mere mechanical barrier, the endothelium is currently understood to be a key player in cardiovascular regulation through the synthesis of numerous factors, which maintain the balance between contraction and dilatation, proliferation, and death [2,3]. Endothelial dysfunction refers to the disbalance in the bioavailability of these endothelial factors in favor of contractile and proliferative processes, which ultimately leads to increased vessel tone and remodeling [4,5]. Endothelial dysfunction is also heavily correlated with aging, much like the overall incidence of CVD, with oxidative stress being a well-known major player [6]. Lack of endogenous antioxidants or increased superoxide anion generated in the arterial wall can play a role. Superoxide anion combines with nitric oxide (NO), one of the key vasodilators in vascular homeostasis, reducing its bioavailability and generating peroxynitrite, a highly oxidizing and stable molecule. 

Many plant-derived substances hold intrinsic antioxidant properties, and clinical trials and epidemiological studies have documented the benefits of dietary phytochemicals to attenuate the oxidative stress associated with chronic diseases and aging [7,8,9]. Diet-derived phytochemicals with health-promoting potential must be held to a pharmaceutical standard, with in vitro studies assessing their mechanisms of action, followed by tests in experimental animals, and clinical trials [10]. One of the richest sources of dietary antioxidants is cocoa, containing several phytochemicals, which may interfere in the initiation or progression of chronic diseases [11]. There is evidence that cocoa consumption lowers the risk of CVD and has positive effects on endothelial dysfunction and hypertension [12,13,14]. Cocoa is highly consumed around the world, generating a large amount of waste. In fact, 90% of cocoa fruit’s total weight is discarded [15]. When rejected as waste, the by-products create detrimental economic and environmental complications. Most plant by-products can be used as animal feed; however, the presence of theobromine (TH), a ubiquitous methylxanthine in the cocoa plant, imposes limitations, since it exerts negative effects on animals at high dietary concentrations [16]. Therefore, the biorefinery of cocoa processing residues needs to evolve towards a circular bioeconomy, and the reuse of by-products rich in bioactive components for the pharmaceutical and food industries is gaining interest [17,18]. The cocoa shell, a by-product removed from the cocoa bean during roasting, contains several bioactive compounds, including methylxanthines -TH, caffeine (CAF)- and phenolic compounds [19,20]. We recently obtained an aqueous extract from the cocoa shell (CSE), based on green extraction methods, and high a phenolic content [21]. We have also demonstrated that CSE has potent antioxidant and anti-inflammatory actions in cell culture, improving mitochondrial function in macrophage-conditioned adipocyte cultures [22]. 

Based on our previous demonstration of CSE antioxidant effects on cells, we hypothesize that this extract could exhibit vasodilatory properties and could be a potential food ingredient for cardiovascular health. Our objectives were to assess: (1) the vasoactive properties of CSE and its mechanism of action, and (2) the capacity of CSE and its principal phytochemicals to improve endothelial dysfunction in arteries from aged rats. 

## 2. Materials and Methods

### 2.1. CSE Extraction and Characterization 

The cocoa shell was supplied by Chocolates Santocildes (https://www.chocolatessantocildes.com, Castrocontrigo, León, Spain). A sustainable green aqueous extraction method was used to prepare an extract rich in phenolic compounds from the cocoa shell (CSE) [21]. Once freeze-dried, the extract was dissolved to achieve a 10^−2^ M gallic acid equivalents concentration, based on total phenolic compounds measurement. UPLC-MS/MS analysis revealed 15 phenolic compounds and two methylxanthines, TH and CAF, present in CSE. The main components of interest, due to their previously described antioxidant properties, were TH (26.4 ± 0.2 mg/g extract), CAF (7.3 ± 0.1 mg/g extract), and total phenolic compounds (47.3 ± 2.3 mg/g extract). The primary phenolic compound was protocatechuic acid (PCA) (0.13 mg/g), followed by epicatechin, procyanidin B_2_, and catechin, ranging from 0.034 to 0.038 mg/g. 

### 2.2. Experimental Animals 

Sprague Dawley rats procured from the Animal House facility of the Universidad Autónoma de Madrid were used. The experiments were approved by the Ethics Review Boards of Universidad Autónoma de Madrid (CEI-UAM 96-1776-A286) and the Regional Environment Committee of the Comunidad Autónoma de Madrid (PROEX 04/19).

The cohorts of rats were of either four to five months (adult) or fifteen months of age (old), males and females. Rats were fed with a breeding diet (Euro Rodent Diet 22; 5LF5, Labdiet, Madrid, Spain) containing 55% carbohydrates, 22% protein, 4.4% fat, 4.1% fiber, and 5.4% mineral, being 0.26% sodium. Drinking water was provided ad libitum in all cases. 

On the day of the experiment, the rats were first weighed and killed by exsanguination by cardiac puncture after carbon dioxide-induced hypoxia. Thereafter, the carotid and iliac arteries were immediately dissected and placed in cold Krebs Henseleit solution (KHS) of the following composition (115 mM NaCl, 4.6 mM KCl, 2.5 mM CaCl_2_, 25 mM NaHCO_3_, 1.2 mM KH_2_PO_4_, 1.2 mM MgSO_4_, 0.01 mM EDTA, 11 mM glucose).

### 2.3. Assessment of Vascular Function 

Carotid and iliac arterial segments were studied by isometric tension recording using an organ bath, as previously described [23]. We first analyzed and assessed the effects of CSE in iliac and carotid arteries from female rats. Since the effects of the compounds under study were demonstrated to be similar in both types of arteries, we chose the iliac artery to evaluate the influence of sex and aging. The arteries were cleaned from perivascular fat, cut into individual segments measuring 3 mm each, and mounted in the organ bath chambers containing KHS. The solution was kept at a constant temperature of 37 °C throughout each experiment. A continuous stream of carbogen gas (95% O_2_ and 5% CO_2_) was pumped into the chambers to maintain physiological oxygen levels and appropriate pH (7.3–7.4). Arterial segments were mounted using two iron wires of a 168 μm diameter, each looped into the artery to create tension. One wire was fixed while the second was hung connected to a force transducer and a data registration system to monitor the changes in tension in the artery throughout the experiment (LabChart, AD Instruments, Dunedin, New Zealand). 

Once mounted in the chambers with the two wires looped, arterial segments were adjusted to a tension of 1.5 g (optimal tension, obtained from preliminary experiments) and left for 30–40 min, readjusting tension until stabilization was achieved. After the equilibration period, 120 mM KCl was added to test arterial function, discarding those segments with low KCl responses. Thereafter, to evaluate endothelium-dependent relaxation, 10^−7^ M noradrenaline (NA) was added. This concentration achieved a sufficient and sustained contraction. Once maximal stable contraction was achieved, a concentration–response curve to acetylcholine (ACh, 10^−11^ to 10^−4^ M) was obtained. Segments with maximal ACh relaxation (≥70%) were considered with endothelium intact (+E). To confirm the dependence of endothelium of the vascular effects of the compounds, some segments were depleted from endothelium by gently rubbing the lumen with a cotton thread and confirming the lack of ACh relaxations ≤30% (−E). 

Subsequently, concentration–response curves to CSE (10^−11^ to 10^−4^ M) were tested either under basal conditions or in vessels pre-contracted with 10^−7^ M NA. To assess the endothelial factors implicated in CSE responses, after the CSE concentration–response curve was determined, the segment was preincubated for 20 min with indomethacin 5 × 10^−6^ M to block prostaglandin (PGI) synthesis and, finally with NN-nitro-l-arginine methyl ester (L-NAME) 10^−4^ M to block NO synthesis. This drug was always added at the end of the experiment. Preliminary experiments demonstrated that neither NA contractions nor ACh relaxations were modified in 3 consecutive concentration–response curves (data not shown).

To assess the effect of CSE and its main components (CAF, TH and PCA) on ACh responses, after the first ACh concentration–response curve was determined, the segment was preincubated for 20 min with one of the abovementioned substances at 10^−4^ M and a second ACh concentration–response curve was completed. Preincubation with CAF (but not with CSE, TH or PCA) significantly reduced the contraction induced by 10^−7^ M NA. Therefore, to achieve the same level of precontraction as in the first curve, in the presence of CAF, 10^−6^ M NA was used. This concentration induced a similar level of tone prior to CAF preincubation (Appendix A).

Except for CSE, all drugs and reagents were obtained from Sigma-Aldrich (St. Louis, MO, USA). Stock solutions of drugs and reagents were dissolved in distilled water except for indomethacin, which was dissolved in 5% NaHCO_3_. Dilutions of ACh and NA were prepared in saline-ascorbic solution to avoid oxidation, and the rest of the drugs used (including CSE) were dissolved in saline solution.

### 2.4. Determination of Basal Superoxide Anion Levels 

Basal levels of superoxide anion generated in the arterial wall was assessed dihydroethidium (DHE), as previously described [24]. DHE is a dye that interacts with superoxide anion locally produced by cells in the vascular wall, generating ethidium bromide, a fluorescent compound that intercalates with DNA and can be detected in the nucleus. The experimental protocol was as follows: the arteries were divided into 2 segments of the same length; one was placed in an Eppendorf tube containing 0.1 mL KHS (internal control) and the other in a tube with the compounds under study (CSE, TH, CAF or PCA) prepared in saline solution at a concentration of 10^−4^ M. The tubes were then placed on a water bath at 37 °C with continuous oxygenation using carbogen gas (95% O_2_ and 5% CO_2_) for 30 min. Then, both segments were incubated with 3 × 10^−5^ M DHE (Sigma-Aldrich; St. Louis, MO, USA), prepared in KHS or saline solution with the compound under study (10^−4^ M) in tubes wrapped in aluminum foil to ensure complete absence of light, and under the same conditions with oxygen for another 30 min. Arteries were then transferred to a multi-well plate, washed twice in saline, and fixed with 4% PFA for 1 h. They were then stained for 15 min with the nuclear dye DAPI (1:500 from stock 1 mg/mL (Life Technologies, D1306, Carlsbad, CA, USA), washed 2 times (15 min each), and stored covered with foil paper for confocal microscopy visualization. 

The arterial segment was mounted whole in a well-made of silicon spacers filled with saline solution. A Leica SP2 spectral confocal microscope was used. Experimental (incubated in CSE or its components) and control (incubated in KHS) vessels were studied the same day and visualized under identical conditions of laser intensity, brightness, and contrast levels with a 40× objective with 2× zoom, using the 488 nm/590–620 nm line of the microscope to locate DHE positive nuclei and the 405/410–475 nm line for detection of all nuclei by DAPI.

From each artery, 4 different regions of the adventitial layer were scanned. Superoxide anion can be produced by NADPH oxidase produced by different types of cells in the vascular wall, including those located in the adventitia. We chose the adventitia to quantify positive cells, since it is the outermost layer, where the laser beam can better penetrate, and the nuclei are clearly identifiable for reliable quantification. To avoid any bias, the regions were randomly chosen based on DAPI wavelength. A stack of 6 or 7 serial optical slices (2 µm thick) in each region was captured and stored for analysis with FIJI free software [25]. A macro was designed to automatize quantification. In brief, a threshold level for DHE visualization was established in preliminary experiments, which was fixed for subsequent analysis. In each stack of images, the number of DAPI positive nuclei and DHE positive nuclei (within the selected threshold) were counted, and the relationship was obtained as % of DHE positive nuclei from total nuclei. The average of the 4 regions analyzed in each segment was used for statistical analysis. 

### 2.5. Statistical Analysis

Statistical analysis was performed using GraphPad Prism 8.0 (San Diego, CA, USA). The normality of the variables was analyzed by Shapiro–Wilk test. Variables with a normal distribution were expressed as means ± standard error of the mean (SEM). Otherwise, they were expressed as the median and interquartile range [Q1; Q3]. The EC_50_ values were estimated in each artery segment using the dose–response stimulation logarithm agonist model. The range of fit for the models was 0.83–0.98. Maximal responses, EC_50_ values, and % of DHE positive cells were evaluated by Student’s t or Mann–Whitney tests according to the distribution of the dependent variable. Differences in concentration–response curves were analyzed using 2-way ANOVA, and each figure legend explains the factors used. A *p*-value < 0.05 was considered significant.

## 3. Results

### 3.1. Vasoactive Effects of CSE 

The vascular effects of CSE were first studied in iliac and carotid artery segments from adult female rats. The functionality of the segment was assessed on the basis of the responses to 120 mM KCl. Average KCl contraction was 1.47 ± 0.10 g in iliac arteries (30 segments from eight rats) and 0.81 ± 0.07 g in carotid arteries (24 segments from six rats). 

Under basal conditions, CSE showed almost no response (Appendix A). This effect was not different between +E and −E segments. 

In arteries precontracted with 10^−7^ M NA, CSE showed a concentration–response relaxation in +E segments, in both carotid and iliac arteries (Figure 1A,B). In iliac arteries, some segments exhibited a small contraction at the highest concentration of CSE used. CSE maximal relaxation was significantly smaller compared to ACh response. The EC_50_ values were also significantly lower in CSE compared to ACh (Table 1). In −E segments, CSE vasodilatory responses were absent (Figure 1C,D). These data suggested that CCE exhibits vasodilatory actions through the endothelium, with lower efficacy but higher potency compared to the classical vasodilator ACh.

The effects of CSE were also tested in iliac arteries from adult male rats. Under basal conditions, CSE did not exert major vasoactive effects (data not shown). As observed in females, in segments pre-contracted with NA, CSE induced a vasorelaxation in +E (Figure 2A), and no relaxation in −E vessels (Figure 2B). CSE maximal relaxations were observed at 10^−5^ M, and the 10^−4^ M elicited a slight contractile response in both arteries with and without endothelium. In male arteries, maximal relaxation and EC_50_ values were also significantly lower in CSE compared to ACh (Table 2). 

### 3.2. Mechanism of Action of CSE

In +E carotid and iliac arteries from female rats, preincubation with indomethacin (5 × 10^−6^ M) did not modify CSE responses (Figure 3A,B). The small contraction induced in some vessels at the highest concentration of CSE was not blocked by indomethacin, suggesting it is not related to the release by CSE of a contractile prostanoid. 

L-NAME abolished CSE relaxation in carotid and iliac arteries from females (Figure 3C,D), in iliac arteries from adult male rats, as well as in iliac and carotid arteries from aged female rats (Appendix A). These data indicated that the vasodilatory effect of CSE was mediated by NO and not by PGI.

We evaluated the possibility that CSE increased the bioavailability of NO through its antioxidant actions. Thus, the capacity of CSE to reduce superoxide anion produced in the vascular wall under basal conditions was assessed by the fluorescent indicator DHE. Arterial segments incubated with CSE (10^−4^ M) exhibited significantly lower numbers of DHE-positive cells than segments incubated in KHS. This effect was observed in both carotid and iliac arteries (Figure 4). 

### 3.3. Effect of CSE on Aged Rats

The effects of CSE and ACh were studied in aged male and female rats and compared to the responses in adult animals. 

In females, no significant differences were observed between adult and old rats in either ACh or CSE responses from the carotid (Figure 5A) or the iliac arteries (Figure 5B). EC_50_ values for ACh or CSE were not statistically different for either type of vessel (Table 3). 

Since iliac and carotid arteries from female rats exhibited similar behavior in terms of ACh and CSE responses, only iliac arteries were studied in male rats. ACh concentration–response curves were smaller in arteries from aged compared to adult males, being maximal responses significantly lower. The opposite trend was observed for CSE, with larger relaxations in iliac arteries from old compared to adult male rats (Figure 5C and Table 4). EC_50_ values were not statistically different between adult and aged males (Table 4).

### 3.4. Effect of the Major Bioactive Components of CSE on Aged Male and Female Rats 

In iliac arteries from aged males, CSE preincubation improved ACh concentration–response curves (Figure 6A). TH and CAF did not modify ACh responses (Figure 6B,C), while PCA tended to increase ACh relaxations, near statistical significance (*p*-value = 0.053) (Figure 6D). In the presence of CSE and PCA, ACh induced vasodilatation at lower concentrations.

In iliac arteries from adult male rats, the effect of CSE on ACh responses was also tested. In segments with large ACh responses (near 100% relaxation), no significant modification was observed, while in segments with poor ACh responses (<30% relaxations), the presence of CSE improved relaxation (Appendix A). 

In iliac arteries from aged female rats with poor endothelial responses, preincubation with CSE (Figure 7A) and PCA (Figure 7D) significantly improved ACh concentration–response curves. However, the responses to ACh were not modified by TH and CAF (Figure 7B,C).

CSE, TH, CAF and PCA reduced the number of DHE-positive cells in iliac arteries from aged male rats (Figure 8).

CSE also reduced the number of DHE-positive cells in iliac arteries from aged female rats (No. cells in KHS = 65.6 ± 6.5; No. cells in CSE = 31.3 ± 10.5; *n* = 5; *p*-value = 0.043). The effects of CAF, TH and PCA were not assessed in arteries from adult rats.

## 4. Discussion

Cocoa contains phytochemicals with proven benefits against hypertension and endothelial dysfunction. Some of these bioactive compounds are also present in non-edible parts of the plant, discarded as waste. The revalorization of these by-products into food ingredients may be beneficial for the environment and human health. We aimed to investigate the potential vasoactive properties of an aqueous extract derived from the cocoa shell, due to its high content in bioactive molecules [21]. Our principal findings were that CSE has vasodilatory properties, irrespective of vessel type, age, or sex, mediated by NO, and exerts its actions at very low concentrations. Regarding the potential mechanism of action, our data suggest that the vasoactive effects of CSE may be related, at least in part, to the antioxidant properties of its main components TH, CAF, and PCA, protecting NO from degradation by superoxide anion. We also demonstrated that CSE also relaxes vessels with endothelial dysfunction, such as in aging, being also able to improve relaxation to other vasodilators. Our results support the possible use of CSE aqueous extract as a food ingredient to improve endothelial dysfunction, also contributing to reducing the environmental problem of food waste in the chocolate manufacturing industry.

Cocoa consumption has been demonstrated to have potential benefits against CVD [12,13,14], attributable to the antioxidant components [26]. We focused our attention on an extract derived from the cocoa shell, considered a food waste with no use, since it contains high concentration of phytochemicals, while in cells, it has demonstrated anti-inflammatory and ROS scavenging capacities [22]. We analyzed the possible vasoactive effects of CSE in rat carotid and iliac arteries and evaluated the effect in males and females of different ages, as a first step towards the development of a nutraceutical with potential cardiovascular benefits. Under basal conditions, CSE did not exert vasoactive actions. However, it exhibited vasorelaxant properties in precontracted arteries, as observed for other established vasodilators, such as ACh, bradykinin, among others. This effect was detected in iliac and carotid arteries from females, both from adult and aged animals. Since the effect was similar in both types of vessels, only iliac arteries in male rats were further studied. CSE showed similar patterns of responses in both sexes in all vessels studied. Compared to the classical vasodilator ACh, CSE exhibited smaller maximal relaxation, indicating lower efficacy. However, EC_50_ values were also significantly lower, suggesting that CSE has a higher potency, and that this extract can exert vasodilatory actions at very low concentrations. The fact that CSE was able to dilate vessels in the nanomolar range, is a relevant aspect, since these concentrations are likely to reach plasma if this extract is used as a food ingredient. 

Regarding the mechanism of action of CSE, we demonstrated that the extract acted in an endothelium-dependent fashion, regardless of sex and artery type. In conduit and muscular vessels, like the carotid and iliac arteries used in our study, the main vasodilators are NO and prostacyclin [27,28,29]. We evaluated whether these endothelial factors contributed to CSE responses using L-NAME, a specific eNOS inhibitor, and indomethacin, a cyclo-oxygenase inhibitor [30]. L-NAME effectively blocked CSE-induced vasodilatation, while indomethacin did not modify the response, indicating that the vasorelaxant properties are tied to NO. 

NO is a short-lived molecule, and its bioavailability depends on the balance between synthesis by eNOS and destruction by ROS. NO is degraded by superoxide anion [31], which is produced in the vascular wall through several enzymatic systems, mainly NADPH oxidases. We have previously demonstrated the presence of NADPH oxidase in the vascular wall of systemic arteries and pulmonary arteries and the elimination of NO by locally produced superoxide anion [32,33]. Based on the previously demonstrated antioxidant actions of CSE in cell cultures [34], we assessed the capacity of the extract to scavenge basal superoxide anion, using the fluorescent indicator DHE in intact arteries. Our results indicate the capacity of CSE to reduce superoxide anion levels. We suggest that this antioxidant effect can contribute to its vasodilatory actions, protecting NO from degradation by superoxide anion, as we have previously described [24,33]. This interaction is likely to be produced by NO diffusion to the sites of superoxide anion synthesis, since superoxide anion is a very unstable radical with a shorter half-life and more limited diffusion across membranes than NO, depending on anion channels. Thus, peroxynitrite formation and damaging effects are likely to occur close to superoxide anion generation sites. In this sense, we showed, in an animal model of hypertension, that increased adventitial NADPH oxidase associates with remodeling and fibrosis of this layer [32]. 

Our data suggest that the effects of CSE are likely related to the antioxidant capacities of its main components TH, CAF, and PCA, previously reported [35], and also confirmed in our study by their ability to reduce superoxide anion levels in vessels from males and females of different ages. The other side of the equation affecting NO bioavailability is its production by eNOS. It has been reported that CAF stimulates endothelial NO synthesis through increased intracellular calcium [36]. In vivo, TH degrades into similar methylxanthines as CAF and, therefore, could also exert this effect [37]. Additionally, phenolic compounds present in CSE, such as PCA, have been reported to increase eNOS expression [38,39]. We cannot discount that, under our experimental conditions, eNOS activity may have been stimulated, contributing to an increase in NO bioavailability. On the other hand, it is not likely that the vasodilatory actions of CSE can be explained by an increase in eNOS expression, since its effect was achieved in a very short time frame (<1 min). However, it would be possible that CSE given as a dietary supplement may be able to stimulate the expression of eNOS or antioxidant enzymes, as previously described for phenolic compounds [11], which deserves further studies. 

CSE vasodilatory effects were similar in different vessels from males and females of different ages, suggesting a similar mechanism and global relaxant capacity in the rat vasculature. Regarding sex differences, we detected that 10^−4^ M CSE induced a slight contractile effect in adult males, less marked in females. This small contraction observed at high concentrations was not blocked by indomethacin, suggesting that it is not produced by the release of a contractile prostanoid. In male arteries, the contractile effect was also observed in vessels without endothelium, which suggests some interaction of CSE compounds with smooth muscle contractile machinery with a sexually dimorphic response. This aspect was not further explored and deserves future attention. 

Once we had demonstrated the vasodilatory effects of CSE, we explored its capacity to improve relaxation in vessels from aged rats, with a compromised endothelial function and oxidative/nitrosative stress [40]. Iliac arteries from 15-month-old males exhibited blunted responses to ACh, as previously evidenced in the aorta from aged rats [41]. However, vasodilatory responses were not affected by aging in female rats, demonstrated in the iliac and carotid arteries. This sexual dimorphism in vascular function has been reported previously; males exhibit decreased NO production and higher levels of ROS compared to females [42]; animal models of hypertension exhibit a higher incidence in males, and oxidative stress seems to play a more prominent role in vascular dysfunction in males than in females [43]. Despite the lower ACh responses in aged males compared to adult rats, there was evidence of greater vasodilatation induced by CSE. To explain these results, we propose that higher levels of ROS in male arteries are blunted by CSE, exerting a larger protective effect on basal NO, increasing is bioavailability. 

We also tested the capacity of CSE to modify the responses to other vasodilators (ACh). Pre-incubation with CSE, improved ACh-induced relaxations in arteries from aged males with endothelial dysfunction, and also in segments with poor endothelium (due to damage during mounting). These results can be explained by the capacity of CSE to improve NO bioavailability, likely through its antioxidant actions, which may be relevant in situations of impaired endothelial function. We tested which of the three main components of CSE were able to contribute to this effect. All of them exhibited the capacity to reduce superoxide anion levels. However, regarding the capacity to improve vasodilatory responses to ACh, only PCA showed a significant action in aged females and near significance in aged males. This could be explained by a synergistic effect of all the CSE components. Additionally, it is also possible that there may be other mechanisms besides their antioxidant actions, which deserves further analysis, particularly regarding PCA and other phenolic compounds present in the extract. 

CSE, TH or PCA did not exhibit effects on NA contractions. However, the presence of CAF blunted it. This effect can be explained by several mechanisms. In vascular tissue, NA acts primarily through alpha-1 adrenoceptors in smooth muscle cells, leading to the inositol triphosphate (IP3)-mediated release of calcium from the reticulum. CAF contains an adenine ring similar to that of ATP, blocking the ATP binding site on the IP3 receptor. Besides, CAF can also reduce contractile responses as a competitive inhibitor of phosphodiesterase, which will produce an accumulation of cAMP, blocking membrane voltage-dependent Ca^2+^ channels or reducing the number of active interactions between actin and myosin [36]. 

In summary, the present data demonstrate the vasodilatory actions of CSE and its capacity to improve relaxation in vessels with endothelial dysfunction. These results support the potential of CSE as a food ingredient for cardiovascular health, also promoting the revalorization of cocoa industry by-products and contributing to the circular economy. The next step would be to assess the effects of CSE in vivo, by administering the extract as a dietary supplement to experimental animals with endothelial dysfunction. We have recently demonstrated that mice receiving CSE at high doses (2000 mg/kg/day) do not show any toxicity signs [44]. The present work showed that CAF, TH and PCA present in the extract may contribute to its antioxidant properties. A preliminary metabolomic study in plasma from rats administered CSE at 250 mg/kg/day for 4–7 days show that TH is present after 4 days of administration and CAF after 7 days. These data suggest that these methylxanthines can reach plasma and could exert their antioxidant and vasodilatory actions in vivo. Further studies also evaluating the presence in plasma of phenolic compounds are needed to further analyze the potential cardiovascular actions of CSE in vivo. Additionally, it has to be noted that when administered in vivo, some phytochemicals present in CSE are likely to be modified by the gut microbiota, leading to the generation of bioactive metabolites derived from phenolic compounds [45], which may also exert vasoactive properties. Their mechanisms of action have not been delineated yet, and this aspect is worth further investigation, in order to provide a complete metabolomic analysis of plasma and feces. 

## 5. Conclusions

An aqueous extract derived from the cocoa shell has vasodilatory properties in different vessels, age ranges, and both sexes.CSE vasodilatory actions are, at least in part, related to its antioxidant properties, protecting NO from superoxide anion degradation.CSE has a higher potency compared to classical endothelium-dependent vasodilators, relaxing arteries at very low concentrations.CSE vasodilatory actions are more prominent in arteries with endothelial dysfunction, also improving vasodilatation induced by other agents such as acetylcholine.Our results indicate the possible use of this extract as a dietary supplement to improve endothelial dysfunction targeting aging and CVD.

## Figures and Tables

**Figure 1 antioxidants-11-00429-f001:**
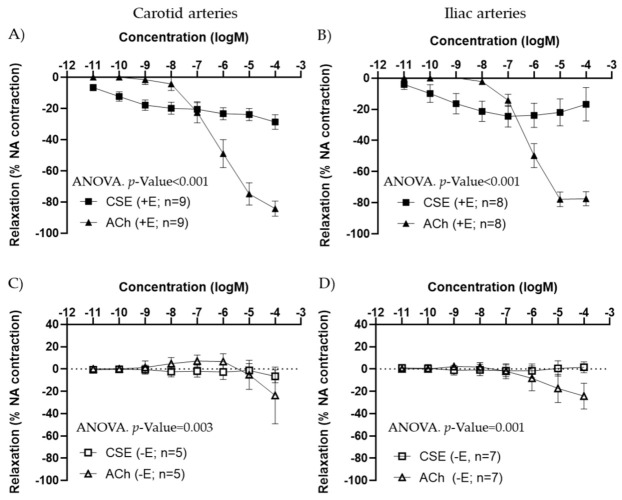
Concentration–response curves to ACh and CSE in precontracted arteries from female adult rats. Data show the effect in carotid (**A**,**C**) and iliac (**B**,**D**) artery segments with endothelium (+E, black symbols) and without endothelium (−E, white symbols). Relaxation is expressed as a percentage of maximal contraction to NA (10^−7^ M). Data represent the mean ± SEM; *n* indicates the number of segments from 4–5 different rats per group. Statistical analysis was performed by 2-way ANOVA, considering concentration and drugs (ACh or CSE) as factors. The *p*-value was extracted of the interaction term of the ANOVA. ACh, Acetylcholine; CSE, cocoa shell extract.

**Figure 2 antioxidants-11-00429-f002:**
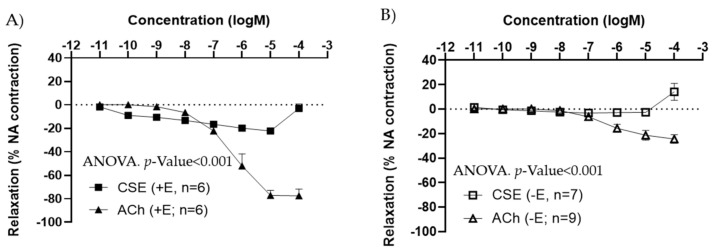
Concentration–response curves to ACh and CSE in precontracted iliac arteries from male adult rats with endothelium (+E, black symbols) (**A**), and without endothelium (−E, white symbols) (**B**). Relaxation is expressed as a percentage of maximal contraction to NA (10^−7^ M). Data represent the mean ± SEM; *n* indicates the number of segments from 3 different rats per group. Statistical analysis was performed by 2-way ANOVA considering concentration and drugs (ACh or CSE) as factors. The *p*-value was extracted of the interaction term of the ANOVA. ACh, Acetylcholine; CSE, cocoa shell extract.

**Figure 3 antioxidants-11-00429-f003:**
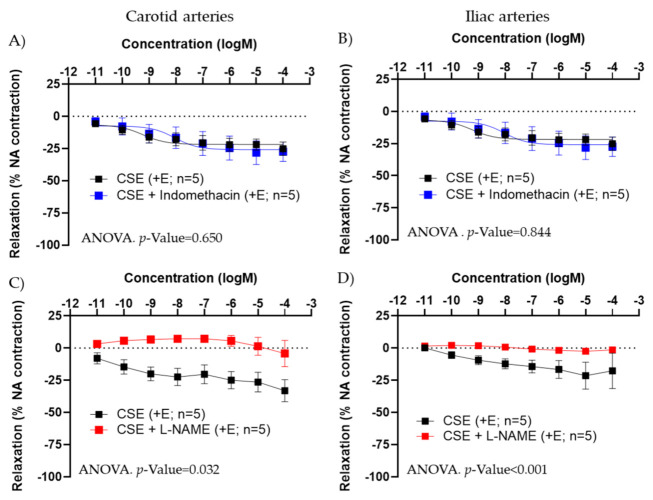
Effect of indomethacin (**A**,**B**) and L-NAME (**C**,**D**) on CSE relaxations in precontracted carotid and iliac artery segments from female adult rats. Relaxation is expressed as percentage of maximal contraction with NA (10^−7^ M). Data represent the mean ± SEM; *n* indicates the number of segments from 3–4 different rats per group. Statistical analysis was per-formed by 2-way-ANOVA considering concentration and drugs (CSE or CSE + Indomethacin/+L-NAME) as factors. The *p*-value was extracted of the interaction term of the ANOVA. CSE; cocoa shell extract; +E, endothelium positive segments.

**Figure 4 antioxidants-11-00429-f004:**
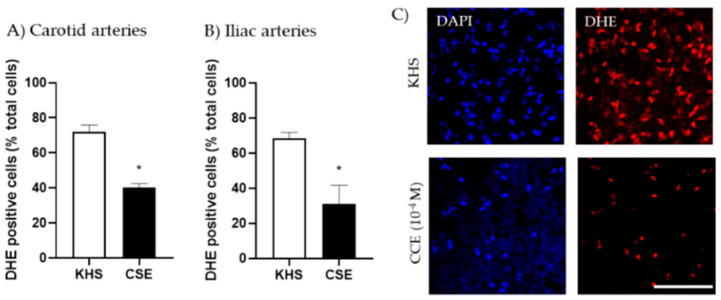
Effect of CSE on superoxide anion levels. Data show % of DHE-positive cells in the adventitial layer from the carotid (**A**) and iliac (**B**) arteries from adult female rats incubated in KHS or CSE (10^−4^ M). Bar graphs represent the mean ± SEM of 5 segments from 5 different rats per group. Data were analyzed by paired Student´s *t* test; * *p*-value < 0.05. Representative confocal images (**C**) of carotid artery segments stained with DHE and DAPI. Images are reconstructions from a 10-μm-thick section of the adventitia taken with a 40× objective with 2× zoom. Image size 512 × 521 pixels. Scale bar represents 25 μm. CSE, cocoa shell extract; DHE, dihydroethidium, KHS, Krebs Henseleit Solution.

**Figure 5 antioxidants-11-00429-f005:**
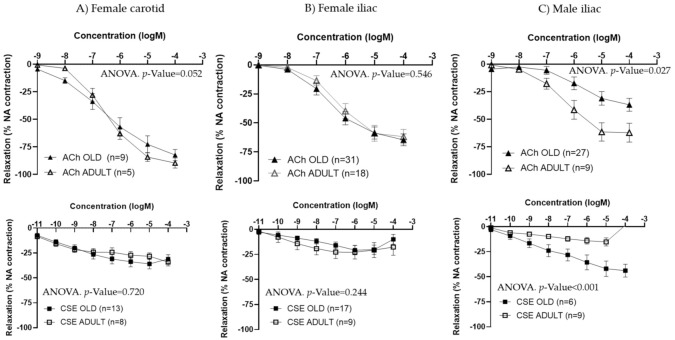
Effect of aging on ACh and CSE responses. Data show ACh and CSE concentration–response curves in carotid and iliac arteries from adult and old female rats (**A**,**B**) and in iliac arteries from adult and old male rats (**C**). Relaxation responses were expressed as percentages of maximal contraction with NA (10^−7^ M). Data represent the mean ± SEM; *n* indicates the number of segments from 3–4 different rats per group; statistical analysis was performed by 2-way-ANOVA, including concentration and age as factors. The *p*-value was extracted of the interaction term of the ANOVA. ACh, Acetylcholine; CSE; cocoa shell extract.

**Figure 6 antioxidants-11-00429-f006:**
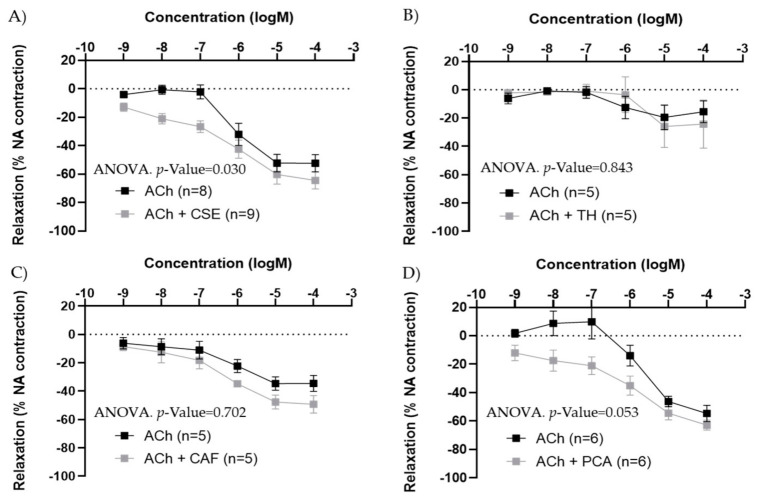
Effect of pre-incubation for 20 min with 10^−4^ M CSE (**A**), theobromine (**B**), caffeine (**C**), and protocatechuic acid (**D**) on ACh relaxations in iliac arteries from aged male rats. Data show ACh concentration–response curves before and after pre-incubation with the compounds. Relaxation responses were expressed as percentages of maximal contraction with NA (10^−7^ M or 10^−6^ M in caffeine-preincubated segments). Data represent the mean ± SEM; *n* indicates the number of segments from 3–4 different rats per group. Statistical analysis was performed by 2-way-ANOVA, considering concentration and components (ACh or ACh +CSE/+TH/+CAF/+PCA) as factors. The *p*-value was extracted of the interaction term of the ANOVA; ACh, Acetylcholine; CSE, cocoa shell extract; TH, theobromine; CAF, caffeine; PCA, protocatechuic acid.

**Figure 7 antioxidants-11-00429-f007:**
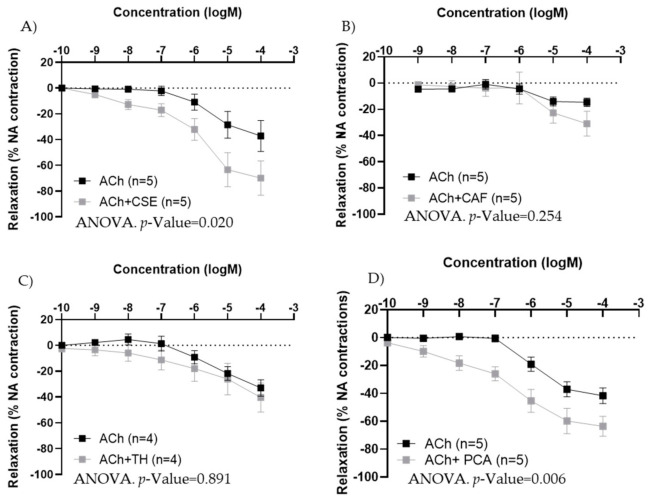
Effect of pre-incubation for 20 min with 10^−4^ M CSE (**A**), theobromine (**B**), caffeine (**C**), and protocatechuic acid (**D**) on ACh relaxations in iliac arteries from aged female rats. Data show ACh concentration–response curves before and after pre-incubation with the compounds. Relaxation responses were expressed as percentages of maximal contraction with NA (10^−7^ M or 10^−6^ M in caffeine-preincubated segments). Data represent the mean ± SEM; *n* indicates the number of segments from 3–4 different rats per group. Statistical analysis was performed by 2-way-ANOVA, considering concentration and components (ACh or ACh +CSE/+TH/+CAF/+PCA) as factors. The *p*-value was extracted of the interaction term of the ANOVA; ACh, Acetylcholine; CSE, cocoa shell extract; TH, theobromine; CAF, caffeine; PCA, protocatechuic acid.

**Figure 8 antioxidants-11-00429-f008:**
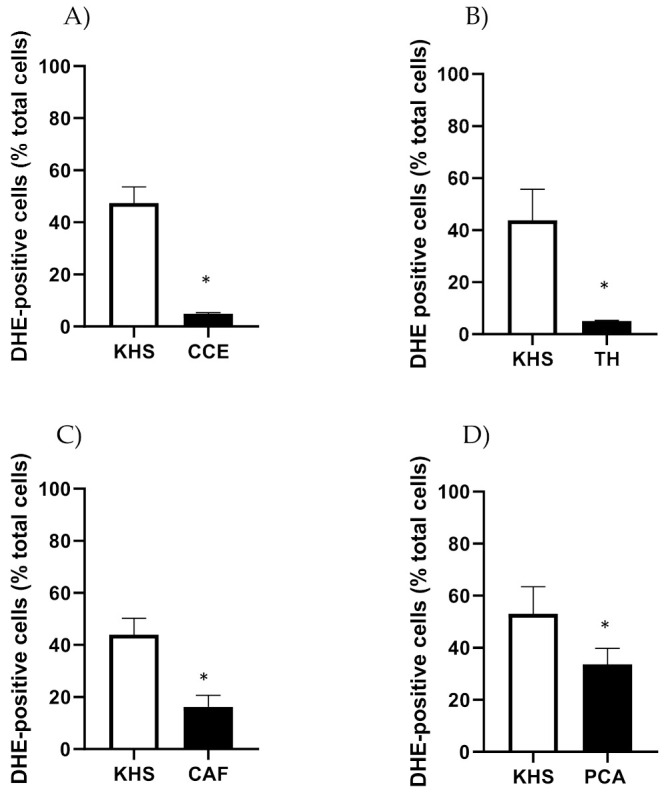
Effect of CSE (**A**), TH (**B**), CAF (**C**) and PCA (**D**) on superoxide anion levels in iliac arteries from aged male rats. Data show % of DHE-positive cells in the adventitial layer incubated with Krebs Henseleit Solution (KHS) or 10^−4^ M of each compound. Bar graphs represent the mean ± SEM of 4 segments from 4–5 different rats per group. Data were analyzed by Student’s *t* test; * *p*-value < 0.05. DHE, dihydroethidium; CSE, cocoa shell extract, TH, theobromine; CAF, caffeine; PCA, protocatechuic acid.

**Table 1 antioxidants-11-00429-t001:** Comparison of EC_50_ values and maximal relaxation to ACh and CSE in carotid and iliac artery segments with endothelium from female adult rats.

Female Artery	EC_50_ ACh	EC_50_ CSE	*p*-Value	Max ACh (%)	Max CSE (%)	*p*-Value
Carotid (*n* = 9)	7.46 × 10^−7^ [5.24 × 10^−7^; 3.13 × 10^−6^]	7.26 × 10^−10^ [1.98 × 10^−10^; 1.07 × 10^−7^]	<0.001 ^a^	−69.71 ± 4.38	−23.90 ± 3.92	<0.001 ^b^
Iliac (*n* = 8)	5.51 × 10^−7^ [1.96 × 10^−7^; 1.48 × 10^−6^]	4.16 × 10^−9^ [4.16 × 10^−10^; 2.44 × 10^−6^]	0.045 ^a^	−74.46 ± 4.79	−21.92 ± 8.67	<0.001 ^b^

EC_50_ values are shown as the median and interquartile range [Q1; Q3] and maximum relaxation (Max) as mean ± SEM, according to the distribution. The *p*-value was extracted by ^a^ Mann–Whitney or ^b^ Student’s *t* tests. ACh, Acetylcholine; CSE, cocoa shell extract; *n* indicates the number of segments.

**Table 2 antioxidants-11-00429-t002:** Comparison of EC_50_ values and maximal relaxation to ACh and CSE in iliac artery segments with endothelium from male adult rats.

Male Artery	EC_50_ ACh	EC_50_ CSE	*p*-Value	Max. Ach (%)	Max. CSE (%)	*p*-Value
Iliac (*n* = 6)	3.04 × 10^−7^ [1.75 × 10^−7^; 1.22 × 10^−6^]	1.20 × 10^−9^ [3.67 × 10^−11^; 9.45 × 10^−9^]	0.002 ^a^	−77.22 ± 4.40	−22.41 ± 2.50	<0.001 ^b^

EC_50_ values are shown as the median and interquartile range [Q1; Q3] and maximum relaxation (Max) as mean ± SEM, according to the distribution. The *p*-value was extracted by ^a^ Mann–Whitney or ^b^ Student’s t tests. ACh, Acetylcholine; CSE, cocoa shell extract; *n* indicates number of segments.

**Table 3 antioxidants-11-00429-t003:** ACh and CSE responses in iliac arteries from old and adult female rats.

Artery	Age	EC_50_ ACh	*p*-Value	Max ACh (%)	*p*-Value
Carotid	Old (*n* = 9)	2.08 × 10^−7^ [1.52 × 10^−7^; 3.75 × 10^−6^]	0.606	−79.34 [−89.55; −58.29]	0.307
	Adult (*n* = 5)	3.31 × 10^−7^ [1.74 × 10^−7^; 7.64 × 10^−7^]		−87.21 [−90.44; −76.51]	
Iliac	Old (*n* = 31)	4.92 × 10^−7^ [8.50 × 10^−8^; 6.82 × 10^−7^]	0.396	−70.62 [−79.65; −33.93]	0.886
	Adult (*n* = 18)	4.93 × 10^−7^ [2.04 × 10^−7^; 2.37 × 10^−6^]		−68.09 [−83.70; −31.95]	
**Artery**	**Age**	**EC_50_ CSE**	** *p* ** **-Value**	**Max CSE (%)**	** *p* ** **-Value**
Carotid	Old (*n* = 13)	9.37 × 10^−10^ [2.87 × 10^−10^; 7.19 × 10^−9^]	0.939	−38.41 [−47.93; −20.23]	0.307
	Adult (*n* = 8)	5.96 × 10^−10^ [1.23 × 10^−10^; 1.23 × 10^−7^]		−25.57 [−32.70; −20.41]	
Iliac	Old (*n* = 15)	2.70 × 10^−9^ [3.47 × 10^−10^; 3.14 × 10^−8^]	0.794	−12.69 [−69.91; −6.04]	0.879
	Adult (*n* = 9)	2.15 × 10^−9^ [2.32 × 10^−10^; 1.63 × 10^−6^]		−16.84 [−41.45; −0.13]	

Data show median and interquartile range [Q1; Q3]. The *p*-value was extracted by Mann–Whitney test. ACh, Acetylcholine; CSE, cocoa shell extract; Max, maximal relaxation; *n* indicates the number of segments.

**Table 4 antioxidants-11-00429-t004:** ACh and CSE responses in iliac arteries from old and adult male rats.

Artery	Age	EC_50_ ACh	*p*-Value	Max ACh (%)	*p*-Value
Iliac	Old (*n* = 27)	7.56 × 10^−7^ [1.16 × 10^−7^; 5.07 × 10^−6^]	0.295	−40.10 [−51.34; −18.00]	0.016
	Adult (*n* = 9)	4.47 × 10^−7^ [1.88 × 10^−7^; 9.13 × 10^−7^]		−71.45 [−84.04; −30.63]	
**Artery**	**Age**	**EC_50_ CSE**	** *p* ** **-Value**	**Max CSE (%)**	** *p* ** **-Value**
Iliac	Old (*n* = 6)	1.20 × 10^−8^ [2.65 × 10^−9^; 5.04 × 10^−7^]	0.228	−36.83 [−64.12; −25.27]	0.005
	Adult (*n* = 8)	2.40 × 10^−9^ [6.02 × 10^−11^; 2.39 × 10^−8^]		−21.05 [−24.67; −3.77]	

Data show median and interquartile range [Q1; Q3]. The *p*-value was extracted by Mann–Whitney test. ACh, Acetylcholine; CSE, cocoa shell extract; Max, maximal relaxation; *n* indicates the number of segments.

## Data Availability

The data presented in this study are available on request from the corresponding author. The availability of the data is restricted to investigators-based in academic institutions.

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
