# Peer review of "Vasoactive Properties of a Cocoa Shell Extract: Mechanism of Action and Effect on Endothelial Dysfunction in Aged Rats"

_antioxidants, 2022, doi:10.3390/antiox11020429_

Round 1

Reviewer 1 Report

Please find attachment.

Author Response

Dear authors,

the manuscript presented is well redacted and shown interesting data about the vasoactive properties of a cocoa shell extract in carotid and iliac arteries from 5-month and 15- 28-month-old rats. However, some issues need to be clarified:

  1. Figure 2 does not show the concentration-response curves to ACh and CSE in pre-contracted carotid arteries of male adult rats. The data should be added in figure 2 and in table 2.

ANSWER. We would like to thank the reviewer for the comments and suggestions. We did not perform experiments in carotid arteries from male rats, since we had previously demonstrated in females that carotid and iliac arteries exhibit the same behavior regarding relaxation to ACh and CSE, both in adult and aged rats (see below data from aged rats).

  1. The results related to the mechanism of action of CSE were conducted in +E carotid and iliac arteries of adult female rats but only in the iliac arteries of adult male rats. Why? Please report the results obtained in the carotids of adult male rats and also of aged rats (male and female) at least as supplementary data.

ANSWER. As indicated above, once we demonstrated that the effects and mechanisms of action of CSE were similar in iliac and carotid arteries from female rats, we did not repeat these experiments in carotid arteries from male rats, since we did not consider it necessary. We think it is important for the readers to know the reason for not performing experiments in both types of arteries and have included an explanation in methods section (lines 120-123). We also provide the data showing that L-NAME abolished the CSE responses in iliac arteries from aged female rats (Supplementary material, Table S1).

  1. The effect of CSE on aged rats was only considered for the iliac artery, why? Please, also report the results obtained on the carotid arteries of the same animals in figure 5 and in table 3.

ANSWER. We did not include the data on the carotid artery since the behavior was the same as the iliac. We have now added the data on ACh and CSE concentration response curves in carotid arteries from adult and aged female rats (new figure 5 and new Table 3 and 4). Since in females the response was similar for both types of arteries, for male rats only iliac arteries were studied.

  1. Again, why the data related to the effect of the major bioactive components of CSE (concentration-response curves to ACh and DHE positive cells) were reported only for the aged male rats? Data should be reported for old female rats and also for adult male and female rats.

ANSWER. We aimed to assess if CSE and its components could improve the effects of a vasodilator (ACh), in the context of endothelial dysfunction. In our experimental conditions endothelial dysfunction, was related to ageing. Therefore, we only tested the effect of preincubation with CSE and its components in aged animals. We agree on the importance to also include data from females and now we have added this information and a new figure (Figure 7). We have also provided data on the effect of CSE on ACh responses in adult male rats (supplementary material, Figure S3), since we think they may be useful for the global interpretation of data.

  1. Finally, data on the effect of CSE as superoxide anion scavenger in the arteries (carotid and iliac) of aged rats (male and female) completely lacking. Please add them.

ANSWER. We have included the data of the effect of CSE on DHE in iliac arteries from aged males to complete Figure 8 (before Figure 7). This omission was a mistake. Therefore, this figure is completed with the effect of CSE and its main components on superoxide anion levels. We have included in text the effects of CSE on iliac arteries from old female rats. Unfortunately, we do not have data on the effect of CSE in arteries from adult males.

The discussion paragraph should be rewritten in light of the missing results. The authors should focus the discussion more on the comparison between adult and old animals and on the differences observed between male and female animals. Finally, the authors must explain why the analysis was specifically performed on the carotids and on the iliac arteries and discuss the data obtained from this point of view as well.

ANSWER. We have expanded the discussion to improve it, in view of the new included results, related to the mentioned aspects, of the effect of sex, age and vessel type.

  1. There are typos throughout the text relating to the abbreviations ACh and CSE.

ANSWER. We have tried to correct these and other typos in the text.

Reviewer 2 Report

The authors investigated the vasoactive actions of cocoa shell extract (CSE), a by-product containing theobromine, caffeine and protocatechuic acid as major phytochemicals. Using carotid and iliac arteries from 5-month and 15–28-month-old rats they showed the vasodilator properties of CSE and its major phytochemicals.  CSE caused endothelium-dependent vasorelaxation regardless of sex, age, or vessel type but blocked by L-NAME. The authors conclude that CSE has vasodilatory properties associated with increased nitric oxide bioavailability, related to its antioxidant phytochemicals, being a potential food ingredient for diseases related to endothelial dysfunction. This is an interesting study but the following points need to be addressed.

Evidence indicates that gut microbes metabolize phytochemicals into phenolic metabolites and metabolites mediate the bioactivities of many of the phytochemicals. In the present study, the authors used cocoa shell extract and phytochemicals to assess their vascular effects. Hence the biological relevance of the study should be discussed.

Did the authors assess endothelial nitric oxide synthase (eNOS)? This will be an interesting endpoint to relate the vasodilatory effects of CSE and its phytochemicals.

Reviewer 3 Report

The purpose of this paper was to examine the influence of cocoa shell extract (CSE) on reactivity of carotid and iliac arteries. The authors report that CSE relaxes arteries and protects against aging-induced alterations in vascular function.

Comments/Concerns:

  1. It appears that CSE can be added to the list of many other antioxidants that protect blood vessels from a variety of disease states. With that said, what is the physiological significance of these data? The authors suggest that supplements can be given (to humans presumably) to help with endothelial dysfunction. How might this occur? The authors found the CSE relaxed vessels, which would decrease blood pressure. In addition, the concentration of CSE given to isolated vessels seems pharmacological and not physiologic. Thus, the overall significance of these data, without giving CSE systemic for a prolonged period of time, seems lost.
  2. What are the novel concepts to be gained by these studies? In other words, other than examining the influence of CSE, what novel mechanistic concepts are gained from these studies?
  3. In Figure 2, the highest concentration of CSE has not influence on vascular diameter, yet in Figure 3 (in females) the same concentration of CSE produces marked relaxation. Why? Further, how can the authors measure the amount of increase in vascular function following CSE when CSE by itself produces marked relaxation in some animals?
  4. Other than staining with DHE, did the authors measure levels of superoxide in vessels from adult and aged (male and female) animals?

Round 2

Reviewer 1 Report

Dear authors,
thank you for taking my suggestions into consideration.
Now the manuscript is better articulated and complete.
In my opinion it is suitable for publication.
I have no further request to suggest.

Author Response

We would like to thank the reviewer for the useful comments and suggestions.

Reviewer 2 Report

The authors addressed the reviewers comments.

Author Response

(The authors gave the same response as above.)

Reviewer 3 Report

The authors have responded to my comments/suggestions, however one is still left with the conclusion that this is just another antioxidant that restores impaired vascular function. There is very limited novel information contained in this study.

Author Response

We want to thank you for the time spent to review our manuscript.

We are sorry about this impression and, of course, we cannot change it. We would like to point out at the novelty to use a food waste product, generated in large quantities and detrimental for the environment, to design a new food ingredient with health promoting properties.

We agree that the techniques provided in the manuscript are not novel. However, they were used to provide an answer regarding the potential vascular effects and mechanism of action of the tested product. We think this work is needed and relevant for the future development of the ingredient and to decide on the possible target population.